

# Monitoring public awareness about the endangered northern bald ibis: a case study involving primary school children as citizen scientists

Didone Frigerio[1,2], Verena Puehringer-Sturmayr[1,2], Brigitte Neuböck-Hubinger[3], Gudrun Gegendorfer[1], Kurt Kotrschal[1,2] and Katharina Hirschenhauser[3]

[1] Core Faciity Konrad Lorenz Forschungsstelle for Behaviour and Cognition, University of Vienna, Grünau im Amtal, Austria
[2] Department of Behavioural Biology, University of Vienna, Vienna, Austria
[3] University College for Education of Upper Austria, Linz, Austria

Corresponding author
Didone Frigerio,
didone.frigerio@univie.ac.at

## ABSTRACT

**Background**. Citizen science has evolved over the past decades by motivating members of the public to interact with scientists and actively participate in scientific research and monitoring. For this purpose, a proficient communication is mandatory in order to efficiently convey messages and reduce the gap of knowledge between scientists and lay people. In the present study, we aimed at evaluating the multiplying effect of children, who were trained to communicate their knowledge on an endangered bird species in order to engage the local community in the long-term ornithological monitoring of the free flying and individually marked colony of northern bald ibis (NBI, *Geronticus eremita*), which was established at the research station in 1997.

**Methods**. Pupils of the local primary schools were in regular contact with researchers, enjoyed outdoor encounters with the birds, and were invited to talk about their experience with as many people as possible. Later on, they acted as surveyors to assess the knowledge of the public on (i) the general knowledge about the species, (ii) specific knowledge about the local colony, and (iii) attitudes towards science. In two different years of evaluation (2012 and 2016) a total number of 387 persons were surveyed. The questions were generated together with the pupils and their teachers and the questionnaires were similar for both years of evaluation. All queries were in a closed format.

**Results**. Our results show an increase in the proportion of correct answers provided by the surveyed persons between the two years of evaluation. Education-based activities may encourage children to effectively act as multipliers of information and attitudes. This has the potential to induce sustainable changes with respect to attitude towards science, at least among local communities. Furthermore, the study suggests caution with the quality of some information reported by citizen scientists, which might be solved by more careful training actions and more specific information about local particularities. Even though the study would have gained more informative power with some additional precautions than in its current form, our findings recommend the empowerment of pupils as multipliers of scientific knowledge.

## INTRODUCTION

Surveys on the public understanding of science in Western communities show that a majority of people cannot perceive the social impact of scientific findings, which unravels a communication gap between scientists and lay persons (i.e., members of the public; *Bensaude-Vincent, 2001*; *Pardo & Calvo, 2002*). As a consequence, communicating science to the general public and responding to public demands about knowledge is increasingly recognized as a responsibility of scientists (*Greenwood, 2001*; *Leshner, 2003*; *Brownell, Price & Steinman, 2013*). The aim to effectively promote levels of general knowledge and science-affinity in society via understanding science is indeed a challenging task (*Racine, Bar-Ilan & Illes, 2005*; *Illes et al., 2010*; *Keehner & Fischer, 2011*). Yet, the effectiveness of educational initiatives strongly depends on the strategies scientists adopt in order to interactively communicate research interests and novel insights to the public (*Van der Sanden & Flipse, 2016*; *Rauchfleisch & Schäfer, 2018*). However, *Binder, Scheufele & Brossard (2012)* suggested that both, low levels of public attention for science and a lack of civic participation in scientific activities can nullify the process of science communication (*Medvecky & Leach, 2017*; *Wilkinson, 2018*). On the other hand the interaction between lay people and scientists, as well as the active participation of people in scientific research are recently gaining attention worldwide (*Shirk et al., 2012*; *Frigerio et al., 2018*) and this indeed benefits science communication (*Constant & Roberts, 2017*).

Over the past decades, citizen science has developed active involvement of interested amateurs in a wide range of disciplines, e.g., by reporting observations, taking measurements or analysing data, at different stages of scientific projects. Evaluation programmes have shown that in addition to the spread of scientific facts and understanding (*Brossard, Lewenstein & Bonney, 2005*; *Trumbull, Bonney & Grudens-Schuck, 2005*), some participants have substantially contributed in developing experimental designs and/or formulating research questions, as well as objective conclusions (*Trumbull et al., 2000*). Involving people from within and outside academia is not entirely new, as for example bird-watching activities may be considered worldwide among the first citizen science programmes (*Irwin, 1995*; *Bonney & La Branche, 2004*). However, education-based activities can be considered a promising emerging field in citizen science (*Wiggins & Crowston, 2011*). They aim especially at increasing scientific literacy and raising awareness about conservation issues (*Bela et al., 2016*; *Bonney et al., 2009*). Such projects are beneficial for all groups of participants and bear a still untapped potential (*Frigerio et al., 2018*). For instance, *Hirschenhauser, Frigerio & Neuböck-Hubinger (2016)* showed that the specific knowledge and interest acquired by pupils during extracurricular science education activities in the frame of citizen science projects were retained over longer periods than usual as compared to the performance of a control group which was not involved in the citizen science activities. In fact, the participatory training effects lasted at least over a period of nine weeks, which

corresponded to the summer holidays break in the country where the investigation was conducted. Thus, citizen science is suitable for education as long as it is complementary to the curriculum proposed by school authorities, as for instance by providing relevant opportunities for the application of previously abstract knowledge to the pupils' lived experiences (e.g., informal education occurring on top of a structured curriculum, *Wilde et al., 2012*; *Rogoff et al., 2016*).

The northern bald ibis (*Geronticus eremita*, NBI) was listed as critically endangered since 1994. Due to the positive development of the last wild population in Morocco, the species was recently downgraded to endangered (*BirdLife International, 2018*). Among several conservation projects started in the past years (*Boehm & Pegoraro, 2011*), a sedentary, free roaming, individually marked and semi-tame NBI-colony was established in 1997 at the Konrad Lorenz Research Station (KLF, Grünau im Almtal, Upper Austria; 47°48′E, 13°56′N) by hand-raising zoo-bred chicks (*Tuckova, Zisser & Kotrschal, 1998*; *Kotrschal, 2007*) in coordination with the European Breeding Programme (*Boehm, 1999*). The KLF still provides important research results about the social behaviour of these colony breeders (*Puehringer-Sturmayr et al., 2018*; *Frigerio et al., 2016*; *Tintner & Kotrschal, 2002*), contributing know-how for different NBI conservation projects in the past years, as for instance the human-led-migration by the Waldrappteam (http://waldrapp.eu; *Fritz & Unsoeld, 2015*) or the reintroduction in Spain (*Quevedo et al., 2004*).

The KLF-flock of NBI is housed in a large breeding aviary, which is kept open year-round, allowing the birds to roam the feeding grounds in the region. During times of frost or snow cover the birds are provided with supplementary food. In the frame of a research project involving several schools of the region (from kindergarten to high school), children acted as knowledge multipliers in order to inform the native people how to participate in the monitoring of the local avian population and to provide information about the sightings of individual NBI on the meadows of the valley. School children were involved in manyfold indoor and outdoor activities (i.e., in the classroom and in the field, see methods section for more details) aiming at fostering their content knowledge on the species. In both years of evaluation, the participating children experienced regular contacts with the researchers and outdoor encounters with the birds, as well as input by visiting professional experts. With the expertise of recognizing juvenile and adult birds, as well as individual ones, the pupils were then encouraged to act as multipliers of knowledge in the village by sharing the information with their families and telling as many persons as possible about their lived experiences. After six months of scientific inputs, the knowledge of the local community on NBI was assessed by involving pupils of the local primary schools as interviewers (paper-pencils questionnaires) with two very similar questionnaires run in two different years, in 2012 (pilot-project) and 2016. The participating children were conducting the surveys face-to-face and later on they were joining the scientists for the analysis and the discussion of the results. We aimed at assessing differences in the knowledge about the local NBI population and gaining experience in targeting educational activities, as well as the quality of the information provided to the local community (science to public and vice-versa). Overall, due to the long-term presence of the research station in the valley, its regular press releases, as well as the resonance of the citizen science activities

with the children (*Frigerio et al., 2012*), we expected people to be aware of NBI research. We expected the children's activities to contribute setting up an emotional relationship between the village and the KLF and fostering interest for its research activities. Results are discussed with respect to lessons learnt and possible improvements.

## METHODS

### Ethical statement

This study complies with all current Austrian laws and regulations concerning working with wildlife. Animal observations were performed under Animal Experiment License Number 66.006/0026-WF/V/3b/2014 by the Austrian Federal Ministry for Science and Research (EU Standard). We confirm that the owner of the land, the Duke of Cumberland, gave permission to conduct behavioral studies on his site. A partnership agreement was signed between the research institution (KLF) and the authorities of both schools joining the project. Furthermore, according to the Austrian privacy policy (BGBI. I Nr. 165/1999), a declaration of consent was signed by the parents of the children participating in the project. The questionnaires received specific ethical approval from the Rector's Office of the University College of Education (30.04.2015). In addition, we asked for consent from the legal school authorities.

The main research interests of the KLF are physiological and cognitive mechanisms underlying social life in avian species. Notwithstanding, the KLF is also engaged for years in promoting young talents and knowledge transfer with the main aim of fostering awareness and gaining acceptance in the local community.

The present study was conducted in two different years: a first time as a pilot-project in 2012 (where only adults were surveyed) and a second time with minor modifications in 2016 (when also children were involved). In both years of evaluation the pupils were actively joining in a research project on NBI. In this context, the researchers offered regular scientific input to the participating classes (at least once per month) for a total of 10 months spread over two school-years (i.e., 5 months per year). At the beginning of the project the researchers (DF, GG) visited the children at school and introduced them to the bird species, its specific behavior and the research questions (*Puehringer-Sturmayr et al., 2018*). The pupils were trained to identify the species, to distinguish the age of the birds (i.e., as juveniles or adults), as well as to spot individuals by the color combination of the rings on their legs (*Frigerio & Gegendorfer, 2013*). Afterwards, the pupils were visiting the Konrad Lorenz Research Station and enjoyed their first close contact with the free-roaming birds, fulfilling the same tasks as at school, i.e., distinguishing individual birds by the colored leg rings and observing their behavior. The pupils also visited the local game park, where they had the opportunity to enter the aviary together with the scientists. Furthermore, they were able to improve their skills in identifying individual birds during an excursion around the area (*Frigerio & Gegendorfer, 2013*). Finally, two international NBI experts visited the schools and presented their work. The participating pupils met the local researchers a total of eight times. In between, they were encouraged to tell as many people as possible about their experiences with the research project. Later on, both the pupils and their teachers

were directly involved in generating the questions for the survey. In 2016 it was generally opted for keeping the 2012-questions and the school classes were asked if they wanted to add further queries.

Therefore, the questionnaires were similar for both years of evaluation; the total number of questions was 14 in 2012 and 15 in 2016. All queries were in a closed format, i.e., a choice among three (2012, "yes", "no", "I do not know") or five (2016, "yes" and "rather yes", as well as "no" and "rather no" and "I do not know") answers was required. "Rather yes" and rather no" were provided to better approach people that could have been unsure about a clear "yes" or "no" answer. Post-hoc, in order to compare the results of the two years of investigation, the answers "yes" and "rather yes", as well as "no" and "rather no" of the 2016-questionnaires were pooled into "yes" and "no" respectively.

In 2016 one open question was added ("How well do you feel informed about the local NBI colony?"). In both years of data collection, the questionnaires had two main foci: (1) general knowledge on NBI biology and (2) specific knowledge on the NBI population of the KLF. In 2016 a third focus was added (i.e., the public attitude toward scientific research (*sensu Jones, Howe & Rua, 2000*)). Questions and correct answers from both years of evaluation, 2012 and 2016, are presented in Table 1. Five of the six questions (1 to 5) had a clear "correct" answer, i.e., "yes" (questions 1, 2 and 5) or "no" (questions 3 and 4). None of the possible answers for question 6 could be assessed as correct or not, as the query dealt with the subjective perception of information on the NBI.

In the two different years of evaluation a total number of 387 persons were surveyed. In 2012 thirteen pupils aged 9 to 10 years of the local primary school (school A) surveyed a total of 67 adults (i.e., older than 18 years; 30 men and 37 women). No additional information about the age of the surveyed people was collected in 2012. In 2016 forty-eight pupils aged between 9 and 10 years of two local primary schools (school A and school B) surveyed a total of 320 people including children in legal school age (i.e., 7 to 16 years old; $n = 156$, 66 boys and 90 girls; mean age $\pm$ SD $= 10.40 \pm 2.75$) and adults (i.e., older than 18 years; $n = 164$, 75 men and 89 women; mean age $\pm$ SD $= 52.92 \pm 17.61$). The children had free choice about how to pick the people for their surveys. Depending on the age of the pupils, some teachers preferred to give the survey as a homework over a longer period of time, whereas others, with younger pupils, chose to use excursions in the village to ask people for answers. The questionnaires were anonymous; however, the children were instructed to make sure that each person was only asked once. For this purpose, they were directly asking the surveyed person if that was his/her first participation in the survey. Therefore, we can exclude that anybody was asked twice, at least within the same year of investigation.

All statistical analyses were done by R version 3.5.1 (*R Core Team, 2018*), using the packages "lme4" (*Bates et al., 2015*) for generalized linear models and "ggplot2" (*Wickham, 2016*) for visualizations. To look for differences between the two years of investigation and to test whether people that felt well informed, were also able to answer more correct questions, we defined the queries 1 to 5 as single response variables and the interaction term query 6 (specific knowledge: I feel well informed about the local NBI colony) and year as fixed factor. Furthermore, to analyse whether the perception of feeling well informed

**Table 1 List of the questions and their relative answers available in the questionnaire in both years of investigation (2012, 2016).** Please note that at the time of data collection NBI was still classified as critically endangered (downgraded to endangered in summer 2018; query 2, focus 1).

| Investigation year questions | 2012 possible answers | 2016 possible answers | 2012 and 2016 correct answer |
|---|---|---|---|
| **Focus 1: general knowledge on NBI** | | | |
| 1. NBI are colony breeders | 1. Yes, no, I do not know | 1. Yes, rather yes, no, rather no, I do not know | 1. Yes |
| 2. NBI are critically endangered | 2. Yes, no, I do not know | 2. Yes, rather yes, no, rather no, I do not know | 2. Yes |
| 3. Adult NBI have feathers on their heads, juveniles not | 3. Yes, no, I do not know | 3. Yes, no, I do not know | 3. Yes |
| **Focus 2: specific knowledge on the local NBI population** | | | |
| 4. The local Almtal NBI colony spends the winter in the South. | 4. Yes, no, I do not know | 4. Yes, rather yes, no, rather no, I do not know | 4. No |
| 5. Birds of the local NBI colony can be easily observed on the meadows around the villages of Gruenau and Scharnstein. | 5. Yes, no, I do not know | 5. Yes, no, I do not know | 5. Yes |
| 6. I feel well informed about the local NBI colony. | 6. Yes, no, I do not know | 6. Yes, rather yes, no, rather no, I do not know | 6. Not applicable |
| **Additional focus 3 (only 2016): attitude towards science** | | | |
| 7. Is science important for the society? | 7. Not applicable | 7. Yes, rather yes, no, rather no | 7. Yes |
| 8. Is conservation important for the society? | 8. Not applicable | 8. Yes, rather yes, no, rather no | 8. Yes |
| 9. Do you trust scientific findings? | 9. Not available. | 9. Yes, rather yes, no, rather no | 9. Not applicable |

changed over the years of evaluation, query 6 was defined as response variable and year as fixed factor. For the analyses we used generalized linear models with a binomial distribution and logit link function. The results are presented as percent of change(delta) of correct, vague and wrong answers between the two years of data collection. This was done within the group of adult people. In the case of the children, who were only surveyed in 2016, results are presented as proportions (percentages) of correct answers for each of the main foci of the questionnaires separately.

## RESULTS

In general, in both investigated foci, i.e., the general and specific knowledge on NBI, the proportion of correct answers was higher in the second year of evaluation as compared to the first year of data collection (Figs. 1 and 2). In 2012, 58.2% of the surveyed adults were aware that NBI are colony breeders; in 2016 a higher proportion of adults (76.3%; Fig. 1A) and 69.7% of the children (surveyed in 2016 only; Fig. 3A) correctly answered this question. The same trend was found for the question about the conservation status (NBI are critically endangered, Fig. 1B): in 2012, 70.2% of the surveyed people were aware of this, and a higher proportion of adults (89%) in 2016, whereas the result for the surveyed children was 77.6%.

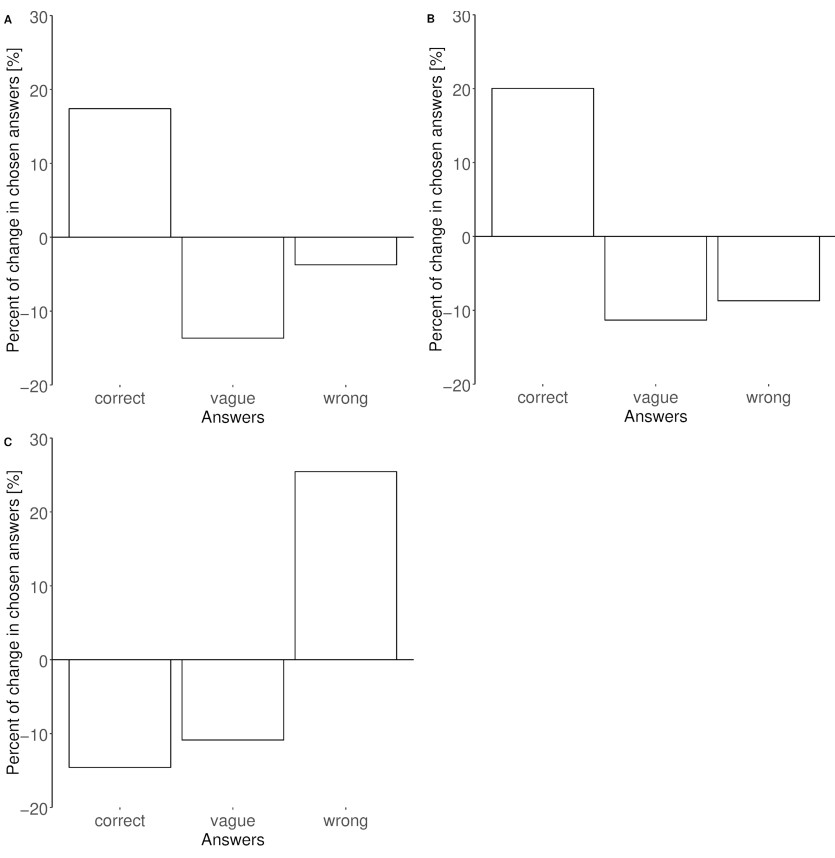

**Figure 1** **Results in percent change of the answers given by adults in both years 2012 and 2016 for the focus 1 "general knowledge on the NBI".** (A) query 1: NBI are colony breeders; (B) query 2: NBI are critically endangered; (C) query 3: adult NBI have feathers on their heads, juveniles do not. Bars underneath the zero line indicate lower rates in 2016 than in 2012; bars above zero line specify higher rates in 2016 than in 2012.

Most of the surveyed people were aware of the possibility of sighting the birds on the meadows of the valley (74.6% in 2012, 91.9% adults and 88.4% children in 2016; Figs. 2B and 3E respectively). Furthermore, in 2012, 43.3% of the people felt well informed about the local NBI free flying colony, whereas in 2016 56.7% adults and 61.2% children felt did so.

We found a trend that more people gave the correct answers in 2016 when they had the feeling to be well informed about the local NBI colony (specific NBI knowledge, query 5). However, the reverse trend was observed when asking people how to distinguish juvenile from adult birds (general NBI knowledge, query 3): in 2012 people were more aware of the morphological differences between adult and juvenile NBI (34.3% in 2012, adults only) than the people surveyed in 2016 (19.6% adults and 22.4% children, Figs. 1C and 3C, Tables 2 and 3). The rates of correct answers for queries 1, 2 and 4 were not significantly different between years, i.e., the variation of the explanatory variables was not better explained by the full model than the null model (Tables 2 and 3). Also, query 4 about the migratory habits of the local colony was answered correctly ("no" in this case) by 22.4%

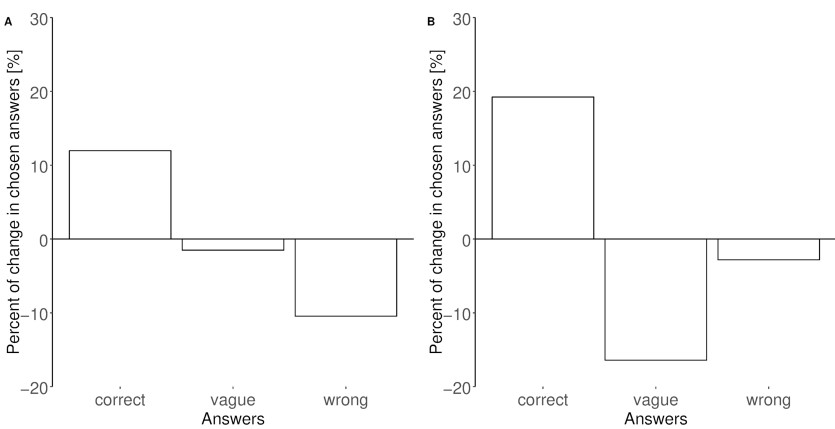

**Figure 2   Percentage change of the answers given by adults in both years of evaluation 2012 and 2016 for the focus 2 "specific knowledge on the local NBI population".** (A) Query 4: the local Almtal NBI colony spends the winter in the South; (B) query 5: birds of the local NBI colony can be easily observed on the meadows around the area. Bars underneath the zero line indicate lower rates in 2016 than in 2012; bars above zero line specify higher rates in 2016 than in 2012.

of adults in 2012, and by 34.7% of the adults and 32.6% of the children in 2016 (Figs. 2A and 3D respectively).

Results related to the third focus are available only for 2016 (Table 4): the majority of people thought that scientific research (94% and 97% of the asked adults and children, respectively), as well as conservation was important for society (99% and 97% of the asked adults and children, respectively). Yet, 13% of the adults and 16% of the school-aged children noted that they were doubtful about the trustworthiness of scientific results.

## DISCUSSION

In the present study, the percentage of correct answers provided in most of the questions asked increased between the two years of evaluation (i.e., over the four study years). Overall in the second year (2016) a smaller proportion of people answered with "I do not know" than in the first year (2012). They rather selected a clear "yes" or "no". Even though we cannot exclude that the children running the survey somehow influenced the answers they received, such evidence might hint to less insecurity by the people when choosing an answer, suggesting that in 2016 people were more confident in their NBI knowledge than in the year 2012.

Overall the findings suggest that the educational activities successfully contribute to a lasting awareness on NBI and scientific research by the local community. Though not directly tested in the present study, the link between correct answers and efficacy of educational activities is plausible. However, adding some more detailed questions in 2016, i.e., in the second year of evaluation, would have increased the informative power of the study considerably. For this purpose, questions like "Did you take part in any of the listed activities…"; "Did you read about the NBI population in the newspaper", or "Did

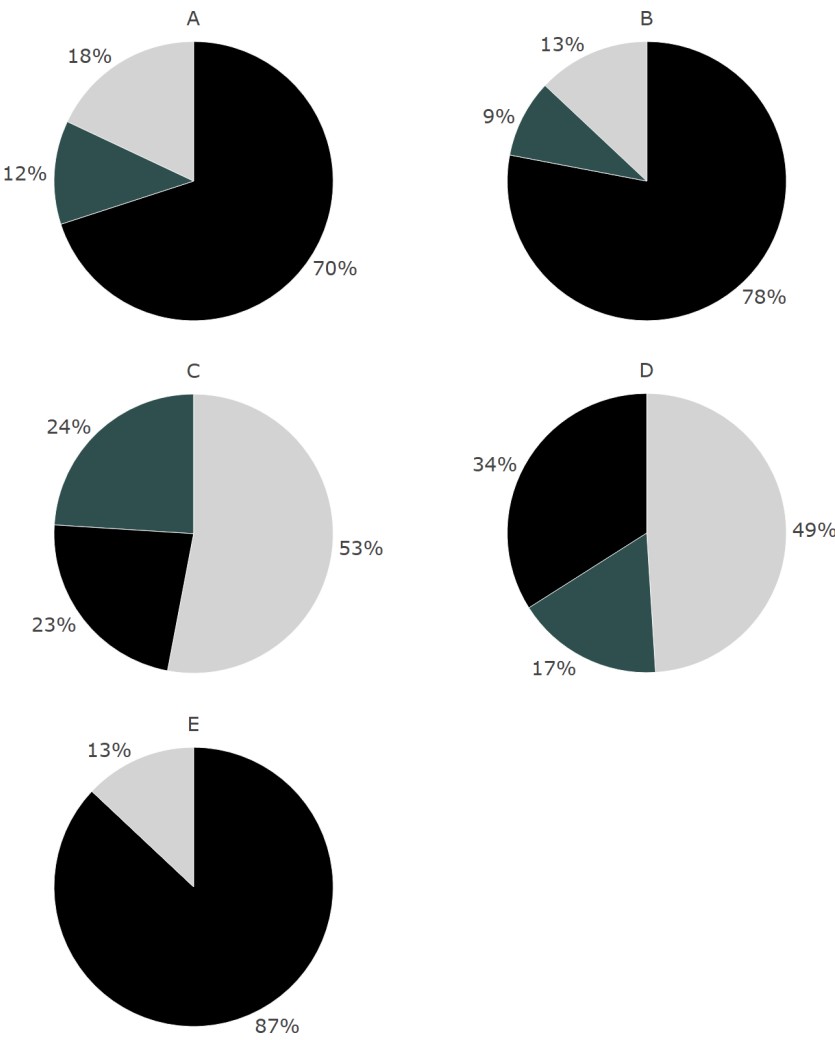

**Figure 3  Proportions of the answers given by children in 2016 about their general (A–C) and specific (D–E) NBI knowledge.** (A) Query 1: NBI are colony breeders; (B) query 2: NBI are critically endangered; (C) query 3: adult NBI have feathers on their heads, juveniles not; (D) query 4: the local Almtal NBI colony spends the winter in the South; (E) query 5: the birds of the local NBI colony can be easily observed on the meadows around the villages of the area. Black = correct answers; grey = vague answers (I do not know); light grey = wrong answers.

you report bird sightings'' would have been helpful for better understanding the patterns observed.

The proposed activities included workshops in the classrooms and hands-on exercises with free ranging birds in the field. Detailed information about the contents of such activities are published elsewhere and included a mix of lecture-style teaching, interactive workshops and hands-on activities (NBI, *Frigerio & Gegendorfer, 2013*; greylag geese, *Frigerio, Hemetsberger & Kotrschal, 2014*; common ravens, *Beck, Frigerio & Loretto, 2016*). Active hands-on involvement may even be more effective than interactive workshops, as recently suggested by *Hesley et al. (2017)* in a programme for reef restoration. There,

**Table 2 Model selection from the generalized linear models. The parameters explaining the response variables (single queries) are provided.**

| Response variable | Model | AIC | *p*-value |
|---|---|---|---|
| Query 1 | Intercept only | 122.675 | |
| | Query 3*year | 125.621 | 0.257 |
| Query2 | Intercept only | 167.389 | |
| | Query 3*year | 167.387 | 0.083 |
| **Query 3** | Intercept only | 315.250 | |
| | **Query 3*year** | **309.179** | **0.006** |
| Query 4 | Intercept only | 310.186 | |
| | Query 3*year | 315.196 | 0.518 |
| **Query 5** | Intercept only | 112.374 | |
| | **Query 3*year** | **110.457** | **0.038** |
| Query 6 | Intercept only | 314.453 | |
| | Year | 312.805 | 0.055 |

participants were trained in reef restoration by unique, experiential learning opportunities to recover degraded corals. Additional surveys among the participants confirmed a significant improvement of their knowledge about coral reef ecology (*Hesley et al., 2017*). Although few studies have rigorously assessed the role of citizen science in changing participants' attitudes, fostering scientific literacy is often considered a ''by-product'' of citizen science activities, (e.g., *Crall et al., 2013*). For example, *Miczajka, Klein & Pufal (2015)* claim that the children involved in a scientific project on seed predation indeed deepened their understanding of the ecology of plant-animal interactions. Furthermore, the participation in such a citizen science project accomplished the requirements of the national school curriculum and provided a valuable opportunity to connect school and community by science learning (*Bouillion & Gomez, 2001*). However, research on learning by experience (i.e., effects of hands-on experiences on students' interests and attitudes) has a long history in the field of science education (*Bergin, 1999*). Even though there is a general consensus that hands-on experiences foster students' learning effectiveness (*Satterthwait, 2010*), there is evidence suggesting that the performance of various hands-on activities can influence students' interest differently (*Holstermann, Grube & Bögeholz, 2010*). Therefore, the positive influence of hands-on activities on students' interest cannot be generalised to every activity, but seems to be rather context-dependent (*Sjøberg & Schreiner, 2005*; *Holstermann, Grube & Bögeholz, 2010*).

Even though the education-based activities had effects on the people's awareness and interest in NBI, some aspects of NBI were not clearly transmitted to the local public knowledge. For example, the percentage of correct answers to the question about the migratory tradition of the local bird population was continuingly rather low (<30%), but still increased between the two years of evaluation (Fig. 2). The NBI flock of the KLF does not migrate. However, the birds spend most of the winter in the aviary of the local game park, where they get fed. Although they could, the birds rarely visit the meadows in the village during the winter months. Therefore, during winter they can be rarely observed

**Table 3  Model coefficients of final models.** The coefficients including standard errors (SE), $z$-value and $p$-value are provided.

| Response variable | Final model | Coefficients | Estimate | SE | z-value | p-value |
|---|---|---|---|---|---|---|
| Query 3 | Query 3*year | Intercept | 0.642 | 0.391 | 1.643 | 0.100 |
| | | query 3 (no) | 0.680 | 0.685 | 0.992 | 0.321 |
| | | query 3 (not sure) | −0.103 | 0.616 | −0.167 | 0.867 |
| | | year (2016) | −0.796 | 0.444 | −1.794 | 0.073 |
| | | query 3 (no)*year (2016) | −0.944 | 0.758 | −1.244 | 0.213 |
| | | query3 (not sure)*year (2016) | NA | NA | NA | NA |
| Query 5 | Query 3*year | Intercept | 2.159 | 0.610 | 3.542 | <0.001 |
| | | query 3 (no) | −0.019 | 0.965 | −0.020 | 0.984 |
| | | query 3 (not sure) | 0.731 | 1.195 | 0.612 | 0.541 |
| | | year (2016) | 2.340 | 1.176 | 1.990 | 0.047 |
| | | query 3 (no)*year (2016) | −2.465 | 1.443 | −1.708 | 0.088 |
| | | query 3 (not sure)*year (2016) | NA | NA | NA | NA |

**Table 4  Proportions of children and adults assigning high importance for science and conservation, and their perceived trust in scientific findings (focus 3 "attitude towards science") in the 2016 survey.**

| Questions | Children (N = 156) | Adults (N = 164) |
|---|---|---|
| Query 7: Is science important for the society? | 95.45 | 93.87 |
| Query 8: Is conservation important for the society? | 94.84 | 98.16 |
| Query 9: Do you trust scientific findings? | 83.23 | 86.59 |

by the local community as free-roaming in the area. Furthermore, the surveyed people probably thought of reports from the human led migration of another NBI colony by the Waldrappteam (http://waldrapp.eu/), an organisation aiming at reintroducing migrating NBI in Europe since 2003 (*Portugal et al., 2014*; *Fritz & Unsoeld, 2015*). This reintroduction project originally developed out of KLF activities, has a strong media presence ever since and may therefore not be distinguished easily from present KLF-related NBI activities by many locals.

According to our study, the surveyed people seem to be reasonably aware of the endangered status of NBI, as well as of the birds' social behaviour (question 1, Figs. 1A–1B). Both aspects might be linked directly to the impact of the project *per se*; however, the latter is indeed a locally relevant topic, as the research of the KLF focuses on investigating NBI social behaviour. Results about how to distinguish between adult and juvenile animals (Fig. 1C) suggest the need for emphasizing observation and ornithological skills in the educational activities. It also suggests caution with data from public sightings which include information on the age of the birds. The sightings are an important part of the long-term monitoring of the KLF and are relevant for generating and answering research questions about habitat use and social behaviour of the colony (e.g., do the birds choose the foraging meadows depending on traditions, i.e., site fidelity, or on habitat quality?). However, mainly because of lacking financial and personal resources this aspect of NBI's behavioural ecology has not been investigated exhaustively yet and it would be best suitable for a citizen science project

(*Dickinson et al., 2012*). In a follow-up project, an App with childlike design (WaldrApp, https://www.citizen-science.at/aktuelleprojekte/item/310-waldrapp) was launched in the region, in order to systematically collect sightings from citizen scientists. In this case, it is mandatory to upload a picture of the bird, so that the researchers may judge the correctness of the information.

Concern about data reliability remains a major issue in citizen science projects (*Kremen, Ullman & Thorp, 2011*). However, different approaches show that data collected by citizen scientists do not consistently differ from those collected by professional researchers (*Cohn, 2008*; *Schmeller et al., 2009*; *Miczajka, Klein & Pufal, 2015*; *Puehringer-Sturmayr et al., 2018*). Therefore, we are confident that a careful and specific training will improve the reliability of the sightings as suggested by other studies on biodiversity (*Kelemen-Finan, Scheuch & Winter, 2018*).

Encouraging children to act as multipliers may not be sufficient to induce significant attitude changes in the local population. However, there is evidence that detailed extracurricular experiences effectively produced long-term concept changes in pupils participating in the school project opportunities offered by the KLF (*Hirschenhauser, Frigerio & Neuböck-Hubinger, 2016*).

The last question on how well the people felt informed about the local NBI colony (less than 61% positive answers) indicates the need and the wish for more frequent and more targeted communication activities by the scientists (e.g., exhibitions and/or public seminars, science cafes), at least at the local level. In fact, the four years of educational activities did produce some changes in the perception of the topic and in the general awareness about NBI. Still, our results of the 2016 survey also show a considerable proportion of people unwilling to fully trust scientific results. How much this was related to our activities remains unclear, as our focal targets for communication were the pupils in schools as natural multipliers of information. The multiplier role for the pupils was beneficial for the aim of the study, as persons familiar and non-familiar with the surveying children were equally motivated to support them in their task. Therefore, involving pupils as multipliers might be a promising approach to induce the attention of the local population for scientific research in general and for the KLF's research activities specifically. This approach contributed to a peer-to-peer context during the street surveys, as none of the parties was a professional researcher.

On the other hand, the transfer of knowledge from the scientists to the pupils might be perceived as a top-down event aiming at a "training the trainer" effect in order to promote a participatory learning community. Both these aspects might contribute to disentangling the thresholds between critical thinking and general mistrusting in scientific evidence in adult people, whereas children seem to be easier in filling themselves with enthusiasm for a specific subject. Peer-to-peer activities involve sharing of knowledge and foster the awareness of participation and responsibility for one's own and others' learning (i.e., public knowledge, *Boud, Cohen & Sampson, 1999*).

Probably due to the teachers' imperative to keep the surveys simple and age-appropriate as well as to our limited experience when working with pupils, our study design lacks some informative power. Indeed, the questions had to be kept simple for the children.

Additionally, attribution of impact to the effects of particular research activities is always a challenge when multiple sources of influence are invovled. However, in order to better disentangle the effects of the educational activities promoted by the children from the communication initiatives run by the research station (reports in public media, presence in schools), it would have been important to add some questions asking the surveyed people about the sources of their knowledge. In fact, these people might have received information to improve their knowledge elsewhere. Such additional queries would have contributed to understand where from and how this particular community receives information effectively. Another meaningful improvement might have been to include a complementary survey without the children–for example ask the same questions in a postal or online survey (which explicitly asked the respondent whether they have had contact with the children involved in the work). This would have provided some 'control' set of observations for the suggested efficacy of the children as knowledge multipliers, as children were supposed to act as multipliers *before* the surveys. A further improvement would have been to employ additional open format questions to supplement the queries 7 to 9, as for instance 'Why do you (not) trust in scientific findings?', which would enable a better interpretation of the 'yes' and 'no' answers.

Our results suggest a major role for the knowledge multipliers in spreading information and promoting societal changes, although at a small scale. The concept of knowledge multipliers is a research topic also in economics science focusing on the rate of technological change (*Da Silva, 2014*). The same concept can be applied in citizen science: for instance, in the "Austrian roadkill project", *Heigl & Zaller (2014)* showed that the participants monitoring dead animals along the streets could improve their understanding of wildlife and conservation issues. They were also willing to share their awareness with others. Thus, voluntary people acted as knowledge multipliers and thereby contributed to public awareness of the topic of the project and of citizen science in general. Therefore, engaging the public in the scientific process may contribute to recognize the wider contexts of science, including culture and policy (*Bonney et al., 2009*; *Mitchell et al., 2017*).

In sum, our results hint at a pivotal role for research initiatives to communicate to, and also, involve lay persons.

## CONCLUSIONS

Our findings show that education-based activities may encourage children to effectively act as multipliers of information and of attitudes. Potentially, this will induce sustainable changes, e.g., changes of attitudes towards science and trust in scientific results, at least among local communities. Furthermore, the study suggests caution with some of the information reported by citizens, which might be solved by more careful training actions and more specific information about local particularities. We highly recommend long-term engagement with a (regional) community, as this can ensure a sustainable willingness to cooperate. In fact, people need time to become confident with, and identify themselves, with a specific topic, such as for instance the nature of scientific research. Integrating children as researchers to poll adults for their knowledge about a biodiversity topic might

be considered as an educative initiative appropriate for young people's growing request to engage in society and global environmental concerns. Furthermore, it seems important to choose focal issues or relevant questions which are of mutual interest for scientists and local communities or part of their everyday experiences (*Ries & Oberhauser, 2015*). Finally, active participation of citizens in research activities, such as conducting research with children and across generations is mandatory for an effective research communication. On these grounds, education-based activities have the potential to benefit all parties involved, that is research, education and society.

## ACKNOWLEDGEMENTS

We gratefully acknowledge C Baer for inserting the data and J Hemetsberger for technical support. Special thanks to J Sieberer, R Witmann, B Leberbauer and their pupils (school years 2011/2012 and 2015/2016) from the Primary Schools in Grünau im Almtal and Scharnstein Mühldorf for the excellent collaboration.

### Funding

Verena Puehringer-Sturmayr was funded by the Programme Sparkling Science, Project SPA-05/26 and by the Programme Top Citizen Science, Project TCS-02/06. Didone Frigerio was funded by the Programme Sparkling Science, Project SPA-05/26 and by the "Back-to-Research-Grant" of the University of Vienna, Faculty of Life Sciences. Gudrun Gegendorfer was sponsored by the company Mayr Schulmöbel. Permanent support came from the "Verein der Förderer der Konrad Lorenz Forschungsstelle" and the "Herzog von Cumberland Stiftung". The funders had no role in study design, data collection and analysis, decision to publish, or preparation of the manuscript.

### Grant Disclosures

The following grant information was disclosed by the authors:
Programme Sparkling Science: SPA-05/26.
Programme Top Citizen Science: TCS-02/06.
Didone Frigerio was funded by the Programme Sparkling Science: SPA-05/26.
Mayr Schulmöbel.
Verein der Förderer der Konrad Lorenz Forschungsstelle.
Herzog von Cumberland Stiftung.

### Competing Interests

The authors declare there are no competing interests.

### Author Contributions

- Didone Frigerio conceived and designed the experiments, analyzed the data, contributed reagents/materials/analysis tools, prepared figures and/or tables, authored or reviewed drafts of the paper, approved the final draft.

- Verena Puehringer-Sturmayr analyzed the data, prepared figures and/or tables, authored or reviewed drafts of the paper, approved the final draft.
- Brigitte Neuböck-Hubinger and Katharina Hirschenhauser conceived and designed the experiments, authored or reviewed drafts of the paper, approved the final draft.
- Gudrun Gegendorfer performed the experiments, authored or reviewed drafts of the paper, approved the final draft.
- Kurt Kotrschal contributed reagents/materials/analysis tools, authored or reviewed drafts of the paper, approved the final draft.

## Human Ethics

The following information was supplied relating to ethical approvals (i.e., approving body and any reference numbers):

A partnership agreement was signed between the research institution (KLF) and the authorities of both schools joining the project. Furthermore, according to Austrain privacy policy (BGBI. I Nr. 165/1999), a declaration of consent was signed by the parents of the children participating to the project. The questionnaire received specific ethical approval from the Rector's Office of the University College of Education (30.04.2015). Afterwards we additionally asked for consent from the legal school authorities.

## Animal Ethics

The following information was supplied relating to ethical approvals (i.e., approving body and any reference numbers):

This study complies with all current Austrian laws and regulations concerning working with wildlife. Animal observations were performed under Animal Experiment Licence Number 66.006/0026-WF/V/3b/2014 by the Austrian Federal Ministry for Science and Research (EU Standard).

## Field Study Permissions

The following information was supplied relating to field study approvals (i.e., approving body and any reference numbers):

We confirm that the owner of the land, the Duke of Cumberland, gave permission to conduct behavioural studies on his site.

## Data Availability

The raw measurements are available in Data S1 and Data S2.

## Supplemental Information

Supplemental information for this article can be found online at http://dx.doi.org/10.7717/peerj.7569#supplemental-information.

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
