# Peer review of "Monitoring public awareness about the endangered northern bald ibis: a case study involving primary school children as citizen scientists"

_PeerJ, doi:10.7717/peerj.7569_

## Round 0.1 · original submission · Major Revisions

The biggest issue with this paper was that the original design of the experiment was weak, which makes it more difficult to have solid conclusions.

Reviewers 2 and 3 make some really good suggestion on how to improve the paper, despite its limitations. It requires a bit of reframing. The abstract needs to clearly state what this paper does. The intro also needs to give a clear and honest idea of what the paper is about. As reviewer 2 also states, it might be good to provide an 'acknowledgment of these flaws and suggestions about what to do better.' This will be critical showing that there is further work to do on this topic.

Reviewer 1 ·

Basic reporting

The overall English is good, I have a few suggestions to improve the language. For example: Line 33 instead of local population, use local community; Line 66: since you use “as well as”, you can delete “both” in this sentence; Line 68: place “indeed” before “benefits”; Line 132: please replace “since” with “for”. Overall, this sentence should be re-structured since it reads rather awkwardly.

In my opinion, the provided results are not sufficient to properly address the stated hypothesis/research question. The abstract as well as the introduction suggest that educational activities with children as multipliers led to an increase of knowledge on the ibis population in the region. However, this has not been tested; the results are purely answers to the same knowledge questions in 2012 and 2016 and their descriptive comparison.

Experimental design

The research questions were twofold: 1) assess differences in the knowledge about the local ibis population in 2012 and 2014 and 2) evaluate the effectiveness of educational activities. Whereas the first research question was answered, albeit only descriptively and with no statistical analysis, the second question has only been addressed speculatively in the discussion. There was neither a data collection regarding educational activities (i.e. question about participation in activities in the questionnaires) nor a description of any of the activities that were on offer. I therefore think that the methods, subsequent results and their interpretation do not match the research questions of this study. Furthermore, if it was just the first research question (without speculation about the effect of educational activities), I have further suggestions for the analysis:
Line 152: I would call it evaluation instead of investigation.
Line 159: how were children taught to make sure that people were only interviewed once?

Line 162-165: This is just a descriptive analysis of the data, why not use Anova or binomial GLm to test the effect of year on the answers? This way, you can statistically assess, whether there actually is a significant difference between the years. For each question, you could run a model (answer ~ year) and assess the significance.

The results are just descriptive and I think that even with the few, rather knowledge-based questions, more statistically valid results could have been extracted. For example, people were asked if they feel well informed about the local colony – a regression between those answers and correct answers could have been used to assess whether this information helped the participants to correctly answer questions (hypothesis: people that feel well informed are able to answer more correct questions).

Line 186-190: you stated in the methods that a third aspect was added to the questionnaire but will not be used here. Why do you then report results for this?

Validity of the findings

I wrongly assumed (based on the introduction), that the impact of educational activities will be assessed in this paper, tested in questionnaires to evaluate their effect. However, this paper purely focuses on comparing answers to questions in 2012 and 2016 and the educational activities that took place in the four years between are used as speculative explanation for the difference. Overall, the discussion is purely based on speculation since none of the links between answers and educational activities were actually tested in the present study. Even though these links are plausible and those educational activities most likely had an impact, this has not been tested. To test these, the addition of a few more questions in 2016 would have been very helpful (for example, questions like “Did you take part in any of the listed activities…”; did you read about the NBI population in the newspaper; did you report bird sightings…). Without these questions, the questionnaire cannot be used to draw the here presented conclusions.

Line 277-278: The quality of reported data has not been mentioned anywhere in the manuscript (or problems with it). Why does this now appear in the conclusion?

Additional comments

Unfortunately, this study cannot deliver what the abstract promises - an analysis of the effect that educational activities with children as multipliers of knowledge have on the local knowledge about the resident ibis population. The only data that was collected where responses to knowledge questions in two different year and the differences between the years was described descriptively. The authors did not actually collect data to address their proposed questions and there was hardly any information and the type and amount of educational activities and how children actually functioned as multipliers of knowledge. The only link between the educational activities and knowledge change is purely based on speculation.

Reviewer 2 ·

Basic reporting

The paper is interesting and well-written and my recommendation is that should be published with some amendments (subject to editor decision based on my comments below).

The case for doing citizen science and including children as knowledge multipliers is well made.
- the structure of the written paper is sound
- Figures and tables are adequate
- data provided is sufficient to support what is written (nowithstanding some of the issues outlined below)
- The referencing is generally ok in terms of the areas covered. There are some places where further literature could be used e.g. lines 86 – 89 could do with some reference to the education literature of in-filed or informal education complementarity to lived experience/informal education. Also there are many references which could have been used to support better design of the survey instrument (see comments in Experimental Design). Note: I have not been through and checked every reference I assume that you are asking for review based on technical expertise not my ability to cross-check in text citations and references which is an administrative task

Experimental design

The survey instrument is quite weak, which limits the analysis and makes the conclusions more speculative than they might have been with some minor changes to the questionnaire and methodology.

It could be grounds to disqualify the paper, but I prefer that the paper is published with some acknowledgement of these flaws and suggestions about what to do better (rather than rejection of the paper) because there are some useful things to learn from the paper and it would be a pity to lose these learnings by not publishing at all.

I understand that the questions had to be kept simple for the children. Additionally, attribution of impact to particular research activities is always a challenge where there are multiple sources of influence. However I feel that some attempt should have been made in the interviews to ask the interviewees about the sources of their knowledge. At several points in the paper (for example lines 111- 115; 235 – 239) reference is made to other really important ways that the interviewees might have received information to improve their knowledge apart from the children as multipliers. I think that the interview should have been constructed to gather this information as a supplement to Q5 – this would help greatly with not only the attribution, but also with understanding where and how this particular community receives information effectively.

Other improvements might have been to include a complementary survey without the children – for example ask the same questions in a postal survey or on-line survey (which explicitly asked the respondent whether they have had contact with the children involved in the work). This would have provided some ‘control’ set of observations for the influence of the child interviewers as multiplier.

A further improvement would be to have some open ended question to supplement Qs 7, 8, and especially Q9. A simple question like ‘why’ would help greatly for interpretation of the ‘yes’/‘no ‘ answers.

At the very least, the Discussion should explicitly acknowledge that the questionnaire has some design flaws, and also suggest how the questionnaire might be improved in future.

There is good explanation of and compliance with ethics given working with animals and minors.

Validity of the findings

Currently the discussion around linking the children as knowledge multipliers and the tested knowledge in the community make sense, but is speculative only because the survey instrument did not collect sufficient information to give any evidence to strengthen the case. This could be more clearly articulated especially if include discussion on some of the points listed in 2. above

Additional comments

A well done to the authors for being innovative and inclusive in the way that citizen science and conservation are being linked. And well done to the children for participating!

Advice would be in this paper to discuss how to improve the survey instruments, explore the use of some open ended questions (acknowledging that this is cannot be made too complex with children - there could be ways to supplement this information gathering by use of supplementary interviews by adults, or by postal or on line surveys as discussed above). I don't think the issue can be fixed for this piece of work but is definitely worth considering for any future work of this sort. I recommend publishing with clear explanation of the shortcomings because there are useful and novel findings.

Minor edits
Line 71 should be e.g. not i.e.
Line 88 – not sure if ‘regional’ experiences is what is meant by ‘informal education’ – this line needs clarification. Perhaps you mean ‘lived experience’?
Also the sentence in.
Line 105 – manyfold rather than manifold
Line 146 – need additional ) after 200

Reviewer 3 ·

Basic reporting

The manuscript ‘Monitoring public awareness about the critically endangered northern bald ibis: a case study involving primary school children as multipliers of scientific knowledge‘ provides a clear, unambiguous, professional English language used throughout. The literature is well referenced and relevant and the structure seems tob e conform with the Peer J standards. Figures are relevant and in adequate quality but might be a little more labelled and described in detail (s. below). The raw data is supplied. I checked for the authors‘ ethical approval statment and human participant/tissue checks and vertebrate animal usage checks are provided.

Experimental design

It might be helpful to provide some further information about the research design and the connection of the presented study to the general work of the KLF with the focus on beneficials due to integrating the idea of citizen science. More specific:
1) who exactly did why which kind of citizen science?
2) the asked questions in the questionaire - why this choice and not differntiating more the term science or the characteristics of the endangered species and its background?

Validity of the findings

Dependent from further information on the experimental design.

Additional comments

The authors of the manuscript ‘Monitoring public awareness about the critically endangered northern bald ibis: a case study involving primary school children as multipliers of scientific knowledge‘ present a highly innovative study that is of interest in the field of citizen science, conservation and education. They integrated primary school children into a long lasting conservation and reseach project on the endangerd northern bald ibis (NBI) at different stages by involving the general local public.
In my opinion it can be seen as a very innovative apporach integrating children as researchers to ‘interview‘ adults about a biodiversity topic. However due to some ambiguity in the experimental design and the validity of of the findings I would recommend a major revision on the manuscript to clearify the aspects mentioned below.

The heading suggests that the children acted as multipliers of knowledge in the sense of informing others about their work – maybe that was done when children learned about the NBI in school and told their parents – but that can not be understood from the text in the manuscript so far. Maybe the authors could further differentiate or clarify that in the manuscript. Furthermore the authors could show that over a time period of four years the knowledge of the public about the NBI increased – however I am really wondering why the sample was totally changed and the sample size increased more than four times – how can the samples be still compared? How do the authors exclude that the people asked in 2016 did not already know in 2012 or did not increase their knowledge over time? An explicit discussion about this issue would be really important and substantiate as this seem to be the main problem to understand the findings of this study at the moment.
Another issue that needs more explanation is the choice of the questions asked in the questionaire - why did the authors decide for this choice of questions? Is there any point why people could answer more often correctly due to a citizen science participation or was that recorded somehow? Here, I would like to ask the authors to precisely explain in this context who exactly did which kind of citizen science? Why it is a benefit of engaging the public? There are of course many important reasons, but I would ask the authors to mention and better explain some of them.
Another point that might need further explanantion is, why the authors suppose that the public after four years would know more about the endangered species – whether it was because of reading the local news, hearing of the project in the supermarket or because they were part of an active monitoring. Here, the reader definitely needs to have further information about how exactly the children or the public contributed by citizen science and more precisely to the conservation of the endangered NBI.

Some further general recommendations to be checked:
Introduction
I think it is indeed a very good and important approach to introduce the public into the conservation of the NBI and communicating the background of the research. I am sure the authors are very well aware of the needs and benefits of doing that especially at the Konrad Lorenz Forschungsstelle (KLF), so I wonder if they could explain that a little more in the manuscript, additionally to the general monitoring and educational programs at the KLF.

Methods
I am wondering if the asked questions in the questionaire are too marginal to provide satisfying information on question i)-iii) by 50% of guessing propability – maybe the authors could explain a little more why they choose these questions.
I am wondering if question 7-9 seem to be a little blanket, however it might be helpful to explain the choice a little more –e.g. why do the authors not think that asking for science trust or understanding is a little vague in comparison of the big differences in research disciplines. Did the interviewed people knew that children were commissioned by a research institute? How to avoid social desirability of ‘trustwothiness‘? It would be helpful to learn more about the questions whether there was a specific intervention that was supposed to change attitudes to science or more specific the knowledge about the NBI project and background?
Did the interviewers capture any further information about the sample like where interviewed people got there informations from?

Discussion
It would be nice and might be helpful to find the structure of the introduction (i,ii,iii) in the discussion as well.

Here, the authors find some further issues in detail that might be checked to help future readers to better understand the study:
55-58: Maybe the authors could explain shortly the mentioned basic scientific concepts – did they play a role within the project in total, respectively in the communication with the public?
62-64: It might be helpful, if the authors could provide a source of that – which communication strategies?
64-66: Here the authors are talking about ‘several studies suggested by recent studies‘, but only one is cited – maybe some more would be helpful?
84-86: It would be nice if the authors could explain whether there was a control that had shorter training effects and therefore highlighting a possible citizen science benefit? Maybe the authors can provide some more information here.
88: It might be helpful to better understand, if the authors could provide a source of that.
105-106: It would be for a better understanding if the authors could explain which kind of activities were conducted in classrooms and provide some examples and an explanation how and why content knowledge should have increased due to the activities.
107-108: The idea of children as multipliers in the village seem to be very promissing – maybe the authors can explain or give the reader some mind picutres what it means to act as multipliers in the village in the name of NBI research?
110: It might be of interest for the readers what kind of acitivites?
111-112: It would be nice to receive some more informations from the authors about the issue which educational activities were supposed to be for whom.
112-113: Maybe the authors could give a short overview of the way of communication taking place between the KLF and the public.
112-115: I am wondering, if not all questions could possibly be answered after reading one informative press release, so maybe the authors can further explain why knowledge should be dependent from children as multipliers? The NBI colony at the KLF excists since 1997, so maybe people learned by time about the project – why do the authors suggest that citizen science explicitly increased people‘s knowledge here?
115-117: I suppose some more information should be provided by the authors how specifically knowledge was increased?
133-135: I am wondering if it might be better to use the term surveyed or somthing similar as the term interviewed is often connected to an oral survey. In this context maybe the authors could tell a little bit more about the details of the survey itself – was it a paper-pencil questionaire, did the children only collect the data or also help to type out them.
145-146: As mentioned before, it might be helpful when the authors could differentiate the science term – in methods and discussion, or explain why they did not in the questionaire.
154-155: Did the authors gather any information where from people had the information to answer the questions in the questionaire?
155-158: Can the authors surely exclude that the children from the same school did not influence each other by telling the right answers to the questionaire?
158-160: It might be beneficial for the study design if the authors could explain how the interviewed people were picked?
160-161: Even so that this issue was mentioned before – it might be helpful if the authors could explain why they interviewed different people? Where is the additional value of using different years when not comparing the knowledge of the same people?
162-163: It might be helpful for the wide readership to receive like two sentences about explaining the meaning of delta of proportions.
196-197: Sorry for that, but returning to the point oft he experimental design – how can the authors suggest that they would receive more right answers from the people that were interviewed in 2016, when other peoples were asked in 2012?
197-198: I would suggest to be more careful to mention that the findings support the communication policy (which is by the way not mentioned or explained in the manuscript) as the ‘research by the locals‘ is not mentioned or explained in detail in the manuscript nor obviously proved by any specific results? I would like to ask the authors to explain that issue to support a better understanding of that statement.
199-201: It might be helful for the understanding of the findings to know more about what was done in the workshops in the classroom respectively what kind of hand-on exercises were conducted?
202-203: I would highly recommend to search for some more literature here as this research topic has a long history already with more differentiated findings. Maybe the authors could cite some more and more specific studies.
206-207: Maybe the authors could explain what kind of change of attitude towards what they had in mind, here.
216-218: I would be careful to talk from ‚‘people’s knowledge to distinguish‘ as they had a 50% option to chose between correct or wrong with given patterns.
218-219: It might be helful if the authors could explain how this is relevant to the study as it is still not clear which role the public engagement plays in the conservation research mentioned in the study.
237-239: In my opinion this seem to be a very important issue, that should possibly be mentioned and explained earlier by the authors for example in the methods as it helps to better understand the manifold work of the KLF and therewith the here presented study.


Figures and tables

A picture of an adult and a juvenile NBI would be nice to get a better knowledge of the endangered species all the research is about.
Figure 1. The caption is very small – maybe the authors could size it up for a better visibility.
Figure 3. It might be helpful to have the figures out of the circles due to a better visibility.
Table 2. It might be helpful to note the sample size in the description.

---

## Round 0.2 · accepted · Accept

I have evaluated your revisions based on the two previous reviews and believe that you have made the appropriate revisions, addressing the reviewer comments. Congratulations on this publication, it's a good paper.